# Identification of potential biomarkers of leprosy: A study based on GEO datasets

**Qun Zhou**[ID]**, Ping Shi, Wei dong Shi, Jun Gao, Yi chen Wu, Jing Wan, Li li Yan, Yi Zheng**[ID] *

Wuhan Dermatology Prevention Hospital, Wuhan, Hubei, P. R. China

* zheng.yi.1206@gmail.com

**Data Availability Statement:** All relevant data are within the paper and its supporting information files.

**Funding:** LiLi Yan is supported by Medical scientific research project of Wuhan Health

## Abstract

Leprosy has a high rate of cripplehood and lacks available early effective diagnosis methods for prevention and treatment, thus novel effective molecule markers are urgently required. In this study, we conducted bioinformatics analysis with leprosy and normal samples acquired from the GEO database(GSE84893, GSE74481, GSE17763, GSE16844 and GSE443). Through WGCNA analysis, 85 hub genes were screened(GS > 0.7 and MM > 0.8). Through DEG analysis, 82 up-regulated and 3 down-regulated genes were screened(| Log2FC| > 3 and FDR < 0.05). Then 49 intersection genes were considered as crucial and subjected to GO annotation, KEGG pathway and PPI analysis to determine the biological significance in the pathogenesis of leprosy. Finally, we identified a gene-pathway network, suggesting ITK, CD48, IL2RG, CCR5, FGR, JAK3, STAT1, LCK, PTPRC, CXCR4 can be used as biomarkers and these genes are active in 6 immune system pathways, including Chemokine signaling pathway, Th1 and Th2 cell differentiation, Th17 cell differentiation, T cell receptor signaling pathway, Natural killer cell mediated cytotoxicity and Leukocyte trans-endothelial migration. We identified 10 crucial gene markers and related important pathways that acted as essential components in the etiology of leprosy. Our study provides potential targets for diagnostic biomarkers and therapy of leprosy.

## Introduction

Leprosy, also known as Hansen's disease(HD), is a skin-related neglected tropical diseases caused by *Mycobacterium leprae*(*M. leprae*), which mainly impairs skin, eye and peripheral nerves to produce a spectrum of clinical phenotypes and can even cause irreversible physical disabilities such as blindness and limb deformities [1, 2]. Even after treatment, patients still need routine follow-up due to nerve damage caused by inflammation within and around peripheral nerve [3]. Patients with leprosy, who suffer from not only physical pain, but also discrimination and low self-esteem, usually lead a poor quality of life [4]. Leprosy has complex pathogenesis, which is characterized by a prolonged incubation period, insidious onset and chronic course. Despite the significant achievements of the global campaign of multi-drug therapy (MDT) over several decades, leprosy transmission is still active in some communities and new cases continue to emerge worldwide [5]. During 2022, 174,087 new leprosy cases

Commission (grant number WX19D56). The funders had no role in study design, data collection and analysis, decision to publish, or preparation of the manuscript.

**Competing interests:** The authors have declared that no competing interests exist.

were reported globally, represented an increase of 23.8% over that in 2021, adding further to the concerns is 5.5% grade-2 disability rate of all new cases [6]. Based on the immune status of the host, according to Ridley-Jopling immunospectral classification, leprosy can be divided into five categories: tuberculoid (TT), borderline tuberculoid (BT), borderline borderline (BB), borderline lepromatous (BL), and lepromatous (LL) [7]. Patients can develop various inflammatory and pathologic reactions including reversal reaction(RR, also known as R1, i.e. "type 1 reactions") and erythema nodosum leprosum (ENL, also known as R2, i.e. "type 2 reaction"), either spontaneously or during therapy [8, 9].

The diverse clinical manifestations and immunopathogenesis of leprosy are strongly associated with the host's immune response including both innate and adaptive immunity. The innate immune mechanisms are key determinants in leading to different clinical manifestations of leprosy and the initiation of nerve damage. The complement cascade, as a vital component of the innate immune system, has been found to be related to increased leprosy susceptibility [10]. Langerhans cells in leprosy skin lesions express CD1a, which is associated with reactional episodes in leprosy [11]. Different macrophagic populations in host tissue can result in different clinical presentations in leprosy and macrophages play key roles in the pathogenesis of leprosy [12]. M. leprae can reduce the efficiency of dendritic cells in inducing T-cell responses and downregulate Schwann cell lineage genes and reactivate developmental transcription factors, thereby leading to the initiation of neuropathogenesis [13, 14]. The adaptive immune system can determine the type of leprosy, lead to a series of pathological lesions and further aggravate the nerve damage, involving T-helper(Th) cells, regulatory T-cells (Treg), natural killer T-cells(NKT), memory T-cells(Tmem), cytotoxic T-cells(Tcyt), antibody-producing plasma cells(CD138), regulatory B-cells(Breg), and memory B-cells (Bmem) [15]. According to previous researches, the course of leprosy is regulated by various complex immune cells and factors. However, the role of immune genes on leprosy molecular pathogenesis and how they interact with each other are largely unknown. Through network analysis, key genes and their interactions in the pathogenesis of disease can be identified. Therefore, we explored the immune-related genes and pathways and revealed their complex interaction network, which can help us better understand the pathogenesis of leprosy.

High throughput microarray platforms can be used to detect gene alterations of diseases and thus discover biomarkers [16]. We provided sufficient samples by integrating multiple microarray datasets to offer more convincing results. Based on crucial genes that were both hub genes of WGCNA and differentially expressed genes, we performed a series of analyses including functional enrichment analysis and protein-protein interaction analysis. Finally, we identified some new biomarkers and used multipartite networks to reveal the interconnectivity between them and their involved immune system pathways, providing novel insights that will help understand the molecular mechanism of this serious disease.

## Materials and methods

### Microarray data from GEO data repository

Gene Expression Omnibus (GEO), as the largest available public microarray database of NCBI, was thoroughly searched for all datasets involving studies of leprosy. Data were retained for further analysis only if they met the following criteria: (1) The study type was limited to expression profiling by array. (2) The sample was from human skin lesion or normal skin. (3) Information about the technology and platform of the study was provided. (4) The study was published publicly and accessible. Finally, microarray datasets GSE84893, GSE74481, GSE17763, GSE16844 and GSE443 were included in our study, containing 130 samples in total (121 leprosy samples vs. 9 normal samples). These 121 leprosy samples consisted of 10 TT, 24

**Table 1. Details of leprosy microarray datasets from GEO database.**

| GSE | Publication | Platform | Classification |
|---|---|---|---|
| GSE84893 | JCI Insight | Affymetrix Human Genome U133 Plus 2.0 Array | ENL:6 |
| GSE74481 | Front Genet | Agilent-028004 SurePrint G3 Human GE 8x60K Microarray (Probe Name Version) | TT:10, BT:10, BB:10, BL:10, LL:4, R1:14, R2:9, CC:9 |
| GSE17763 | Cell Host Microbe | Affymetrix Human Genome U133 Plus 2.0 Array | LL:7, BT:10, RR:7 |
| GSE16844 | J Infect Dis | Affymetrix Human Genome U133 Plus 2.0 Array | ENL:6, LL:7 |
| GSE443 | Science | Affymetrix Human Genome U95 Version 2 Array | LL:6, BT:5 |

TT, tuberculoid; LL, lepromatous; BB, borderline-borderline; BL, borderline-lepromatous; BT, borderline-tuberculoid; ENL, erythema nodosum leprosum, also known as R2(type 2 reaction); RR, reversal reaction, also known as R1(type 1 reactions); CC, normal sample.

LL, 10 BB, 10 BL, 25 BT, 21 RR and 21 ENL, which were involved in all disease types of leprosy so as to avoid generating less reliable results. Details of samples in these datasets that we used for following analysis are provided in **Table 1**.

## Preprocessing of raw data

The selected five gene expression profiles were merged into one file, and log and baseline transformation were done, so as to get rid of potential heterogeneity. We then eliminated the inter-batch differences with R package "sva" [17] and used the default parameters for batch normalization analysis, resulting that a normalized gene expression profile containing data from the five different datasets was obtained for WGCNA and DEG analysis. The normalized gene expression profile can be found in **S1 Table**.

## WGCNA analysis

Weighted Gene Co-expression Network Analysis(WGCNA) was constructed on the normalized gene expression profiles with R package "WGCNA" [18]. Automatic network construction was carried out with soft-thresholding power as 7, minimum module size as 30 and dendrogram cut height as 0.25. Genes in the same module often share a higher level of co-expression. Then we picked out the module which contained genes particularly associated with leprosy by the correlation between modules and clinical traits. In addition, in order to screen out the hub genes to leprosy, we calculated gene significance (GS) to measure the correlation between genes and modules and module membership (MM) to measure the correlation between genes and clinical traits.

## DEG analysis

The normalized gene expression profile containing data from the five different datasets was obtained for DEG(differentially expressed gene) analysis with R package "limma" [19]. We used the default parameters of limma to perform DEG analysis. Then we used heatmap and volcano plot to display differentially expressed gene levels. The heatmap and volcano plot were drawn with R software. To present chromosomal locations of differentially expressed genes, circus was used [20].

## GO annotation and KEGG pathway

To obtain the biological attributes and functional pathways of intersection genes of WGCNA and DEG analysis, Gene Ontology(GO) and Kyoto Encyclopedia of Genes and Genomes

(KEGG) pathway enrichment analyses were performed with R package "clusterProfiler" [21]. Significance was set at $P < 0.01$.

## PPI analysis

Protein-protein interaction(PPI) analysis was carried out with the following databases: STRING [22], BioGrid [23], OmniPath [24], InWeb_IM [25] using Metascape(http://metascape.org). Molecular Complex Detection (MCODE) algorithm [26] was further applied to identify densely connected network components if the network contains more than three proteins.

# Results

## Workflow

Our workflow of bioinformatics analysis is illustrated in **Fig 1**. We obtained 4925 genes in common after preprocessing of raw data downloaded from GEO database. Then we conducted WGCNA analysis and 85 hub genes were screened with the threshold at GS > 0.7 and MM > 0.8. Furthermore, after DEG analysis, 85 differential genes were screened with the threshold at |Log2FC| > 3 and FDR < 0.05, including 82 up-regulated genes and 3 down-regulated genes. The intersection of these two results indicated that 49 genes were crucial and warranted further research. Then these 49 genes were subjected to GO annotations, KEGG pathways and PPI analysis to determine their biological significance in the pathogenesis of leprosy.

## WGCNA analysis

Soft-thresholding power was seted at 7 to construct a scale-free network using pickSoftThreshold function, when the scale independence exceeded 0.9 for the first time($R^2 = 0.906$) and had a relatively high mean connectivity (**Fig 2A**). We then detected gene modules based on the TOM matrix with soft-thresholding power as 7. As a result, eight modules were identified. **Fig 2B** showed the relationships between the identified module genes, indicating that the gene expression was relatively independent among modules. The blue module had the highest correlation with leprosy (cor = 0.74 and $P$ = 2e-23, **Fig 2C and 2D**) among the eight modules, thus we selected the MEblue-grade block for subsequent analysis. The blue module contained 1025 genes, then using GS > 0.7 and MM > 0.8 as cut-off criteria, 85 hub genes were identified. **S2 Table** illustrated the detailed information of WGCNA result including gene names contained in all modules and their GS and MM values.

## DEG analysis

We found 82 up-regulated genes and 3 down-regulated genes after DEG analysis (|Log2FC| > 3 and FDR < 0.05, **Fig 3A**). More information including the fold change and FDR of these 85 genes were shown in **S3 Table**. Heatmap of these DEGs were demonstrated in **Fig 3C**. Chromosome location distribution revealed that chromosomes 1 contained the greatest number of dysregulated genes (**Fig 3B**). Interestingly, while four genes on the X chromosome showed dysregulation (SASH3, CYBB, IL2RG and SH2D1A), none Y chromosome gene was affected.

## GO annotation and KEGG pathway

There were 49 crucial genes both in result of WGCNA and DEG analysis (**S1 Fig**). GO annotation and KEGG pathway were then performed to explore the potential biological functions of these genes. As **Fig 4A–4C**, the GO annotation results showed that the crucial genes were

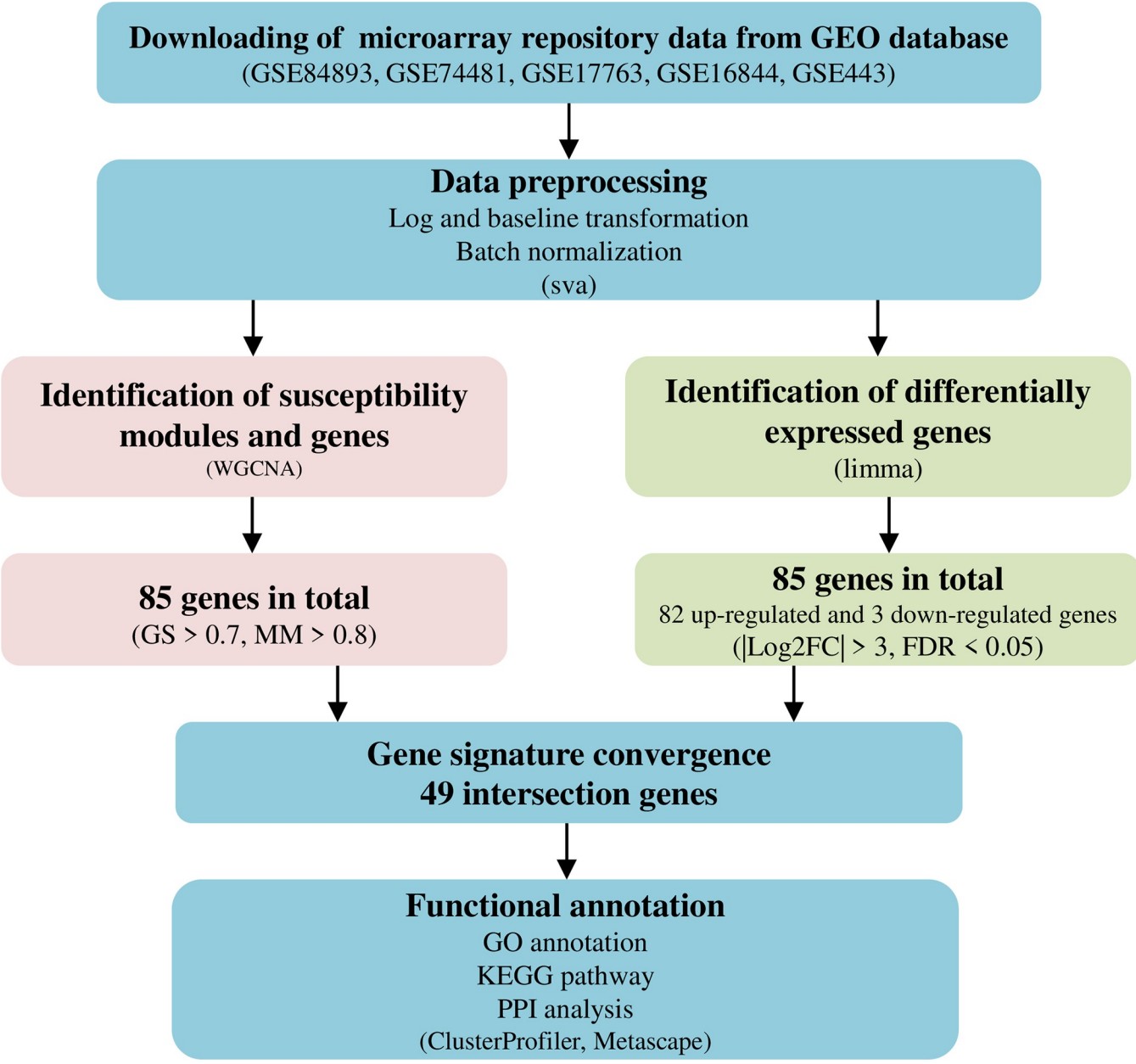

**Fig 1. Workflow of bioinformatics analysis.**

mainly associated with T cell activation, positive regulation of cell activation and positive regulation of cytokine production regarding the biological process. For cellular component, the genes were mainly associated with external side of plasma membrane, secretory granule membrane, cytoplasmic side of plasma membrane. For molecular function, the genes were mainly associated with immune receptor activity, cytokine binding and non-membrane spanning protein tyrosine kinase activity. The KEGG pathway analysis revealed that the crucial genes were predominantly enriched in Chemokine signaling pathway, Th1 and Th2 cell differentiation, Th17 cell differentiation and Natural killer cell mediated cytotoxicity (**Fig 4D**). The complete results of GO and KEGG analyses can be found in **S4 Table**.

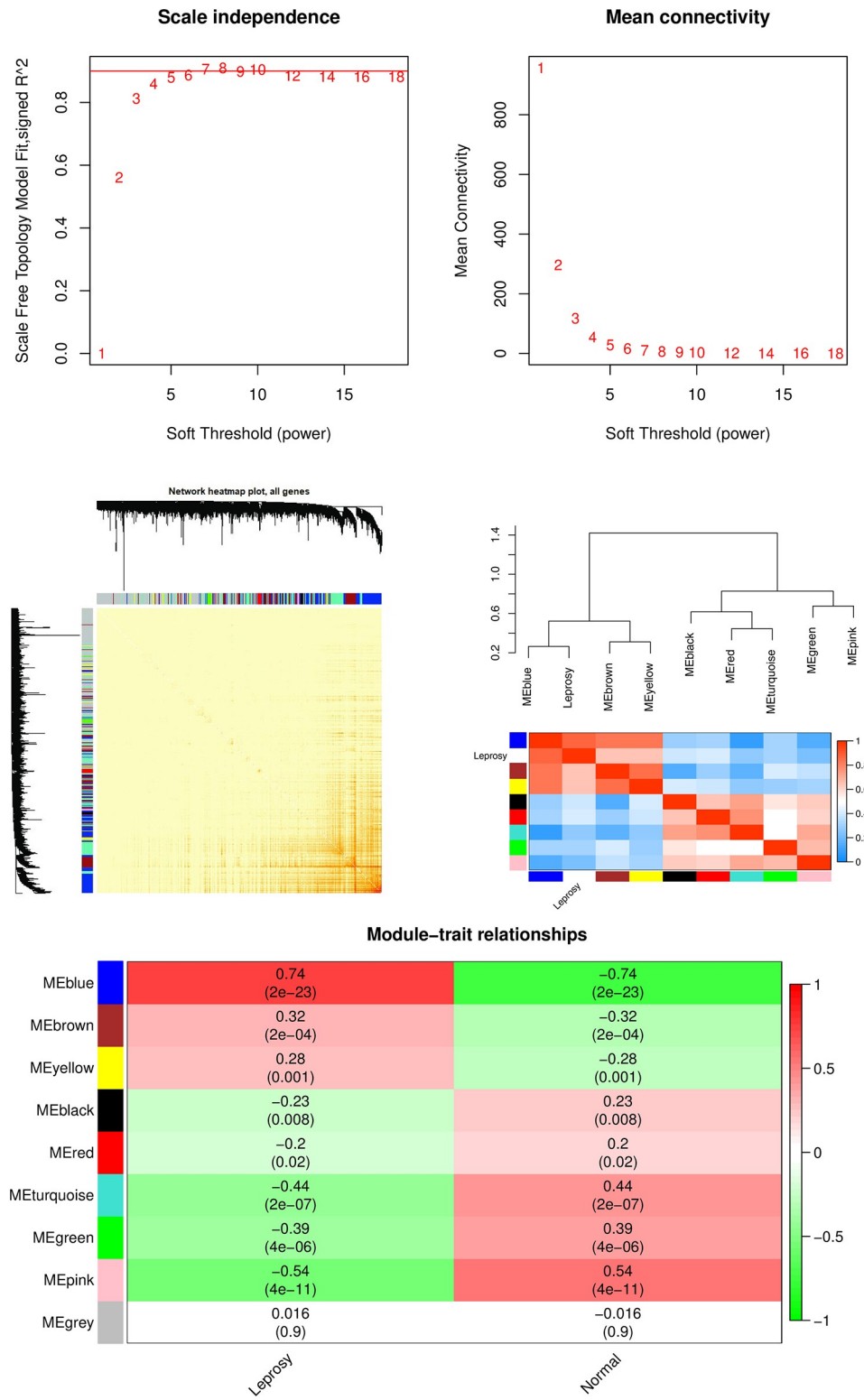

**Fig 2. WGCNA result.** A) Obtaining soft-thresholding power by analyzing the scale-free fit index and mean connectivity of network topology. B) Heatmap depicts the Topological Overlap Matrix (TOM) of all genes of the WGCNA network. The darker the color, the higher the overlap. C) Heatmap of module eigengenes and leprosy trait. D) Heatmap of the correlation between module eigengenes and clinical traits. Each row corresponds to a module, and each column corresponds to a trait. Each square is colored according to the corresponding correlation and labels correlation and *P* value.

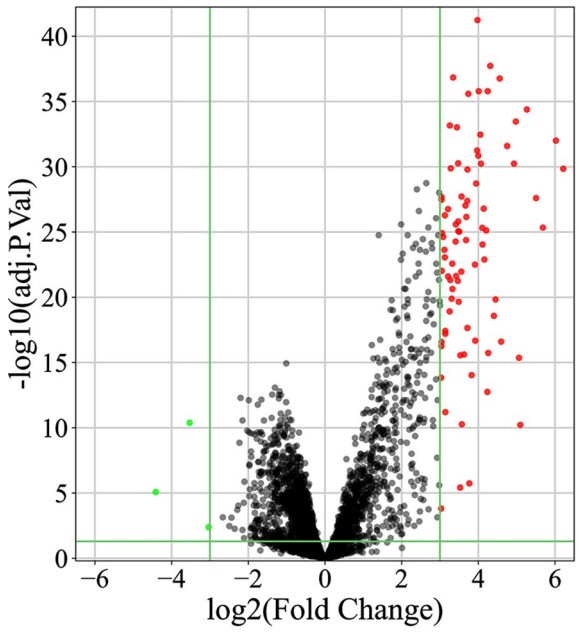

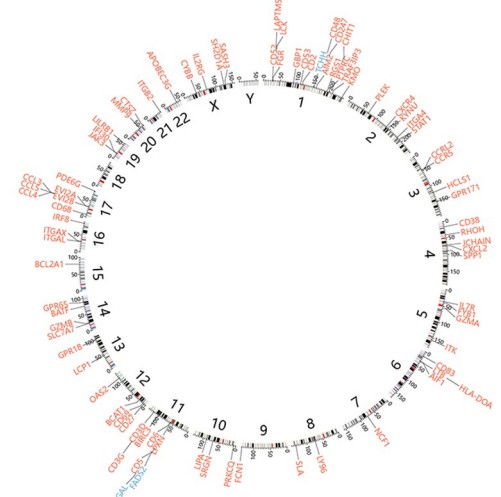

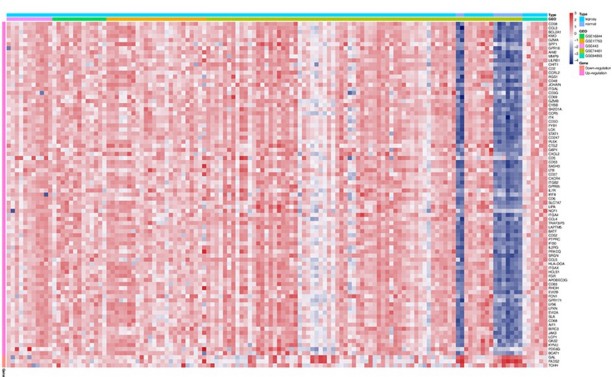

**Fig 3. DEG result.** A) Volcano plot of normalized gene expression profile. B) Chromosome mapping of differentially expressed genes. Red color represents up-regulated genes and blue represents down-regulated. C) Heatmap of differentially expressed genes.

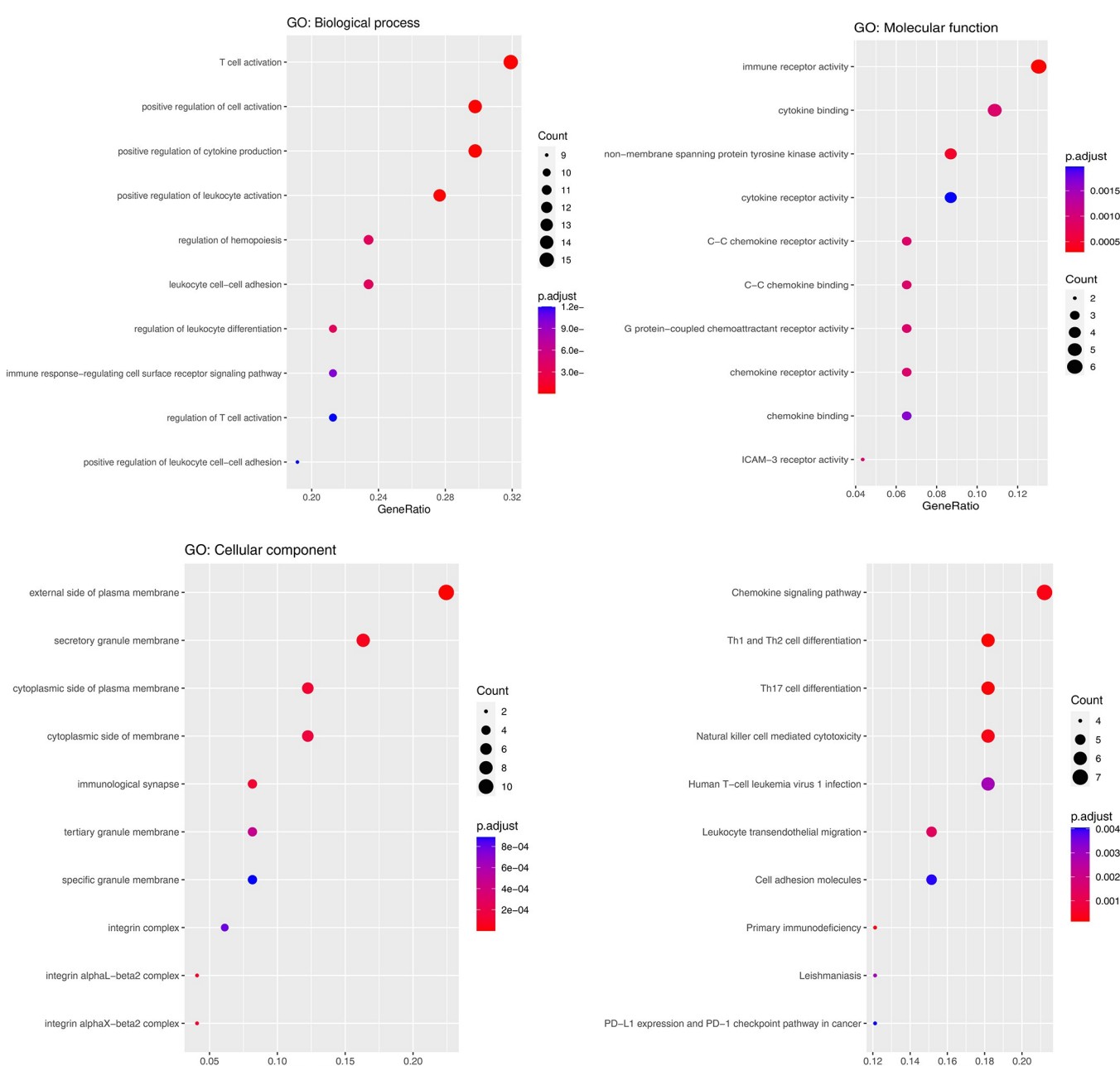

**Fig 4. GO annotation and KEGG pathway of crucial genes related to leprosy.** A) Bubble plots showing GO annotations regarding biological process(BP). B) Bubble plots showing GO annotations regarding cellular component(CC). C) Bubble plots showing GO annotations regarding and molecular function(MF). D) Bubble plots showing KEGG pathway.

## PPI analysis

We also conducted PPI analysis with the intersection genes. The results using MCODE algorithm showed that two components were obtained in which genes can closely interact with each other (**Fig 5A**). The PPI result and MCODE components can be found in **S5 Table**. Then we conducted KEGG enrichment analysis of genes in these components and filtered pathway of p.adjust < 0.05 to draw network with cytoscape, indicating that these genes involved include immune system, immune disease and infectious disease pathways (**S2 Fig**). The whole result of KEGG enrichment analysis of component genes can be found in **S6 Table**. Chemokine signaling pathway contained the largest number of associated genes(JAK3, FGR, ITK, CCR5, STAT1, CXCR4). Next, we filtered the immune system pathways and their related genes. As a result, six pathways connecting 10 genes were finally identified (**Fig 5B**).

## Discussion

As a global disease caused by *M. leprae*, the registered prevalence of leprosy has been decreased substantially from more than 5 million cases in the 1980s to 133,802 cases in 2021. However, there were still 140,594 new cases reported globally in 2021 [27]. Furthermore, leprosy is still a poorly understood illness and considering the disability and dysfunction suffered from this disease, it's worth striving to study the pathogenesis [28]. Varied manifestations of leprosy are associated with the host immune responses to *M. leprae*, involved both innate and acquired immune responses. Many immune cells play important roles in the pathogenesis of leprosy, including macrophages, Schwann cells, dendritic cells, lymphocytes, etc [29, 30]. Therefore, in order to understand the pathogenesis of leprosy, we investigated the immunological pathways and related crucial genes.

Gene expression profiling based on microarray technique has been widely applied in large-scale genomic analysis and biomedical research. Moreover, integrating multiple data can potentially increase statistical power of individual studies [31]. In our present study, we gathered and integrated gene expression profiles from five microarray datasets. Several linkage loci on chromosome 2p14 [32], 6p21 [33], 6q25–26 [33], 10p13 [34], 17q11–q21 [35], and 20p12 [36] may be associated with leprosy susceptibility. The chromosome mapping of differentially expressed genes showed that genes were widely distributed on all chromosomes except Y. By joint analysis of the consolidate data, 49 crucial genes were screened, which both were hub genes of WGCNA(GS > 0.7 and MM > 0.8) and differentially expressed genes($|Log2FC| > 3$ and FDR < 0.05).

We found that these crucial genes were predominantly enriched in T cell activation, positive regulation of cell activation, external side of plasma membrane, secretory granule membrane, immune receptor activity, cytokine binding, Chemokine signaling pathway, Th1 and Th2 cell differentiation, Th17 cell differentiation and Natural killer cell mediated cytotoxicity. The responses of T cells have been proved to be important in determining host immunity and leading to different leprosy development outcomes [37, 38]. Various regulatory T cells, such as Treg and natural killer T cells, can adjust the polarized state of T cell immunity, thus controlling the clinical manifestation [39]. Tuberculoid leprosy is related to Th1 cytokine response, while lepromatous leprosy is associated with Th2 cytokine response [40]. Th17 cells may contribute to lesional inflammation by recruiting neutrophils, activating macrophages and enhancing Th1 effector cells [41, 42]. Cytokines gene polymorphisms play essential roles in shaping the immune responses in leprosy, which even can drive the conversion between functionally antagonistic cells [43]. M. lep*rae* can prevent activation of the host chemotactic response by inhibiting chemokine expression and finally escape destruction by the immune system [44].

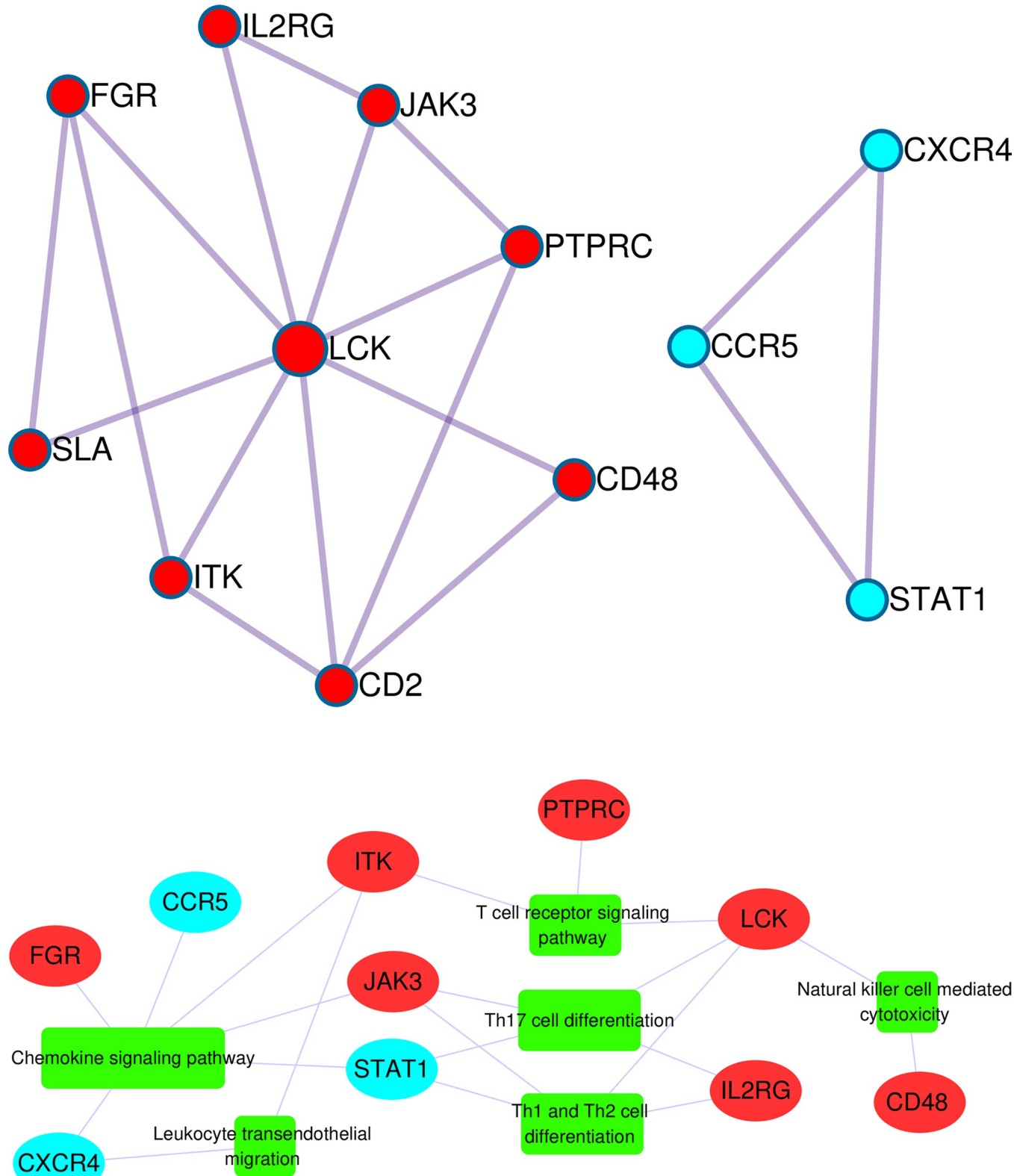

**Fig 5. PPI Analysis of crucial genes related to leprosy.** A) Protein-protein interaction network of two key components identified based on MCODE. Red color represents MCODE1 genes and blue represents MCODE2 genes. B) The network of immune system pathways and component genes. Oval box represents gene and square box represents pathway. The wider the pathway frame, the more genes the pathway contains.

Protein-protein interaction network has displayed the functional connections of crucial genes. Through MCODE algorithm, we identified two densely connected network components. Then we conducted KEGG enrichment analysis of these component genes and extracted the immune system pathways to draw a gene-pathway network, which was composed of 10 genes (ITK, CD48, IL2RG, CCR5, FGR, JAK3, STAT1, LCK, PTPRC, CXCR4) and 6 immune system pathways(Chemokine signaling pathway, Th1 and Th2 cell differentiation, Th17 cell differentiation, T cell receptor signaling pathway, Natural killer cell mediated cytotoxicity, Leukocyte transendothelial migration). Previous studies have proved that these genes play important roles of immunoregulation. ITK signaling is crucial for humoral responses, B cell functions, T cell development and Th2 responses [45, 46]. CD48 is involved in a wide variety of innate and adaptive immune responses, including T cell activation, autoimmunity, granulocyte activity, NK function and antimicrobial immunity [47]. IL2RG plays an essential role in T cells and natural killer (NK) cells production and B cells normal function [48]. CCR5 can regulate IL-2 production and promote T cell proliferation [49]. FGR plays a potential role in FCRL4-mediated immune regulation [50]. JAK/STAT family factors can contain the proliferation of M. leprae by promoting cell-mediated immunity [51]. LCK can determine the T cell signaling via regulating the phosphorylation of various signaling molecules and interact with negative regulators CD45 (PTPRC) leadding to T cell hyporesponsiveness in leprosy progression [52]. CXCR4 may drive the recruitment of lymphocytes to tissue lesions of leprosy patients [53]. Our study had explored the complex relationship between crucial genes and immune system pathways.

Leprosy is closely related to immune response. Most of the damage to leprosy patients is secondary to immunological reactions [54]. Immunological techniques can be very useful in the diagnosis of leprosy, in the follow-up and in detection of relapses [55]. Nutrition status can affect the progress of leprosy through regulating immune pathways [56]. Although there are no useful biomarkers in the clinical setting so far, biomarkers can be used to prevent the spread of leprosy and design interventions to modulate the host's immune response to M. leprae infection and prevent damaging immune-mediated pathologies, which is a focus of future research work [57]. Our study focused on immune markers of leprosy, hoping to be helpful for the diagnosis and treatment of leprosy.

## Conclusion

In summary, we have discovered ten crucial genes(ITK, CD48, IL2RG, CCR5, FGR, JAK3, STAT1, LCK, PTPRC, CXCR4), which may act as potential targets for diagnostic biomarkers and therapy of leprosy. Then we found six related important immune system pathways(Chemokine signaling pathway, Th1 and Th2 cell differentiation, Th17 cell differentiation, T cell receptor signaling pathway, Natural killer cell mediated cytotoxicity, Leukocyte transendothelial migration), and constructed a gene-pathway network to revealed their complex interactions. Our work may improve the understanding of immunological molecular mechanisms underlying the initiation and development of leprosy.

Leprosy still remains endemic within over 140 countries around the world and approximately 200,000 new cases were reported worldwide in 2017. Additionally, leprosy still faces many diagnostic and treatment challenges [58]. As an ancient disabling disease closely related to immunity, we believe that leprosy will eventually be conquered with deeper researches into the potential immune pathogenesis.

## Supporting information

**S1 Table. The normalized gene expression profile of five datasets.**
(XLSX)

**S2 Table. The information of WGCNA module genes.**
(XLSX)

**S3 Table. The information of the 85 differentially expressed genes.**
(XLSX)

**S4 Table. GO and KEGG analysis results.**
(XLSX)

**S5 Table. Protein-protein interaction network and MCODE components identified in the gene lists.**
(XLSX)

**S6 Table. KEGG enrichment analysis of MCODE component genes.**
(XLSX)

**S1 Fig. Venn of crucial genes in result of WGCNA and DEG analysis.**
(TIF)

**S2 Fig. Network of component genes and pathways involved in these genes.** Oval box represents gene and square box represents pathway. Red color represents MCODE1 genes and blue represents MCODE2 genes. The wider the pathway frame, the more genes the pathway contains.
(TIF)

**S1 Graphical abstract.**
(TIF)

**S1 File.**
(R)

## Acknowledgments

We are extremely grateful to all the authors for their hard work to this research.

## Author Contributions

**Data curation:** Qun Zhou.

**Software:** Qun Zhou.

**Visualization:** Qun Zhou.

**Writing – original draft:** Qun Zhou, Ping Shi, Wei dong Shi, Jun Gao, Yi chen Wu, Jing Wan, Li li Yan.

**Writing – review & editing:** Yi Zheng.

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
