## [Decision Letter · Decision Letter 0]

16 Nov 2023

PONE-D-23-26444Identification of crucial genes and functional network features of leprosy based on GEO expression profilesPLOS ONE

Dear Dr. Zheng,

Thank you for submitting your manuscript to PLOS ONE. After careful consideration, we feel that it has merit but does not fully meet PLOS ONE’s publication criteria as it currently stands. Therefore, we invite you to submit a revised version of the manuscript that addresses the points raised during the review process.

Study examines the several crucial gene markers and related important pathways that acted as essential components in the etiology of leprosy, which may enhance our fundamental knowledge of immune molecular mechanisms underlying the initiation and development of leprosy. Leprosy has a high rate of cripplehood and lacks available early effective diagnosis methods for prevention and treatmen .

Article is excellent for the translational perspective, presented in an intelligible fashion and written in standard english however, technical, statistical explanation required extensively for the better understanding of the article. Article should refer latest studies in same field and refer its clinical relevance.

I had appended the reviewer comment for the better understanding and to make article in better shape.

Decision- Major revision with response to all reviewers comment pointwise.

We look forward to receiving your revised manuscript.

Kind regards,

Anshuman Mishra, PhD

Academic Editor

PLOS ONE

Additional Editor Comments:

Study entitled- Identification of crucial genes and functional network features of leprosy based on GEO expression profiles by Y Zheng el al 2023.

Dear Dr. Zheng,

Study examines the several crucial gene markers and related important pathways that acted as essential components in the etiology of leprosy, which may enhance our fundamental knowledge of immune molecular mechanisms underlying the initiation and development of leprosy. Leprosy has a high rate of cripplehood and lacks available early effective diagnosis methods for prevention and treatmen .

Article is excellent for the translational perspective, presented in an intelligible fashion and written in standard english however, technical, statistical explanation required extensively for the better understanding of the article. Article should refer latest studies in same field and refer its clinical relevance.

I had appended the reviewer comment for the better understanding and to make article in better shape.

Decision- Major revision with response to all reviewers comment pointwise.

Reviewers' comments:

Reviewers' comments:

Reviewer's Responses to Questions

**Comments to the Author**

1. Is the manuscript technically sound, and do the data support the conclusions?

Reviewer #1: Yes

Reviewer #2: Yes

Reviewer #3: Yes

2. Has the statistical analysis been performed appropriately and rigorously? 

Reviewer #1: Yes

Reviewer #2: I Don't Know

Reviewer #3: Yes

3. Have the authors made all data underlying the findings in their manuscript fully available?

Reviewer #1: Yes

Reviewer #2: Yes

Reviewer #3: Yes

4. Is the manuscript presented in an intelligible fashion and written in standard English?

Reviewer #1: Yes

Reviewer #2: No

Reviewer #3: Yes

5. Review Comments to the Author

Reviewer #1: This paper presents results from a pipeline of analyses using various R packages. The analyses of publicly available microarray data from GEO repository found 49 crucial genes associated with leprosy susceptibility. These 49 genes were both differentially expressed and hub genes of WGCNA.

For reproducibility's sake, I think, it'd be good if the authors make their R codes used for all analysis publicly available.

Reviewer #2: This manuscript utilizes large-data and bioinformatics techniques to assess relationships and associations between pathophysiology of leprosy and gene/molecular markers. This study aims to provide insight into the pathogenesis of leprosy and identify potential biomarkers that may be implemented as therapeutic targets in the future. The authors implement WCGNA, DEG, and PPI analyses to identify a network of genes predominantly related to immune regulation that may play a role in the pathogenesis and pathophysiology of leprosy.

Major comments:

1. Please explain why the cutoffs and threshold limits (e.g. cutoffs for GS and MM, threshold for WGCNA analysis) were selected, including if similar studies implemented the same cutoffs.

2. Please provide more explanation as to how the statistics were performed beyond the R package(s) utilized. Notably, the supplementary datasets contain statistical values that should be transferred to some capacity into the main text.

3. Verbiage that was used to describe findings should be revised (e.g. revising “we got a gene-pathway network” to “we identified…”; “10 genes were finally got”, etc.)

Minor comments:

1. Please provide more explanation and description of the methods, including how/why each database was utilized.

Overall, this is a study that provides insight into the pathogenesis of leprosy using bioinformatics techniques. However, there are several critical considerations that require attention.

Reviewer #3: Identification of crucial genes and functional network features of leprosy based on GEO expression profiles

Here the authors have collected the bioinformatics analysis with leprosy and normal samples using GEO database. They have screened 85 hub genes using WGCNA. They performed DEG analysis and screened 82 up-regulated and 3 down-regulated genes. Finally, they have identified 10 genes (ITK, CD48, IL2RG, CCR5, FGR, JAK3, STAT1, LCK, PTPRC, CXCR4). They have suggested these 10 genes can be used as leprosy Biomarker because they were active in 6 immune system pathways, including Chemokine signaling pathway, Th1 and Th2 cell differentiation, Th17 cell differentiation, T cell receptor signalling pathway, Natural killer cell mediated cytotoxicity and Leukocyte trans endothelial migration. The research is exciting and valuable enough to be published in PLoS One. Before being accepted for publication, the piece must, however, be appropriately revised in accordance with the remarks offered below.

Comments:

1. The title of the manuscript should be modified.

2. If possible graphical abstract should be provided. According to journal format if it is not possible in the main text, it can be provided in the supplementary file for better understanding of the readers (optional).

3. The Abstract portion should be modified.

4. Line Numbers should be added throughout manuscript.

5. The introduction portion should be rewritten.

6. The author should give a strong hypothesis why this study in details.

7. Introduction section:

(3rd Paragraph) Line no -1: detecte should be corrected.

8. Materials and method section:

Here the authors have simply written they have taken the data and have used the different softwares for the data analysis but the detailed analysis regarding how it is doing that is completely lacking. This should be addressed properly.

9. Results section:

All figures need a better resolution. The author should also mention the image source and the software used to make the images.

10. Conclusion section:

The conclusion portion should be rewritten and the author must add a paragraph on future prospects

11. After the conclusion section there should be some small sections like: Supporting information, Acknowledgments, Author Contributions etc

12. Correct the PubMed ID of Reference No. 40.

Recommendation: Major Revision

6. PLOS authors have the option to publish the peer review history of their article (what does this mean?). If published, this will include your full peer review and any attached files.

Reviewer #1: No

Reviewer #2: No

Reviewer #3: No

---

## [Author Response · Author response to Decision Letter 0]

17 Jan 2024

Dear Editors and Reviewers:

Thank you for your letter and for the reviewers’ comments concerning our manuscript entitled "Identification of crucial genes and functional network features of leprosy based on GEO expression profiles" (Manuscript ID: PONE-D-23-26444). We feel great thanks for your professional review work on our article. As you are concerned, there are several problems that need to be addressed. According to your nice suggestions, we have made extensive corrections to our previous draft, the detailed corrections are listed below.

Response to Reviewer #1: 

Comments:

For reproducibility's sake, I think, it'd be good if the authors make their R codes used for all analysis publicly available.

Response to comment: We thank the reviewer for the kind comments. We will be happy to share the data with readers, and this effort can improve the reproducibility and citation of the article. As the software and its parameters used in the analysis have already been provided in the article, the data result can be easily reproduced. Moreover, we have provided the main codes used in the article as file “Main_workflow.R”. Additionally, the reader can contact the author to access more details.

Response to Reviewer #2: 

Major comments:

1. Please explain why the cutoffs and threshold limits (e.g. cutoffs for GS and MM, threshold for WGCNA analysis) were selected, including if similar studies implemented the same cutoffs.

Response to comment: We thank the reviewer for the kind comments. We selected the cutoffs and threshold limits for their statistical significance. And our selections are evidence based. For example, we conducted WGCNA analysis and screened hub genes with the threshold at GS > 0.7 and MM > 0.8, because we can use gene significance (GS) to measure the correlation between genes and use modules and module membership (MM) to measure the correlation between genes and clinical traits. Weiwei Liang used GS > 0.2 and MM > 0.8 for hub genes(Liang, W., Sun, F., Zhao, Y., et all. Identification of Susceptibility Modules and Genes for Cardiovascular Disease in Diabetic Patients Using WGCNA Analysis. Journal of diabetes research, 2020, 4178639. https://doi.org/10.1155/2020/4178639). We applied a stricter constraint to discover more meaningful genes.

2. Please provide more explanation as to how the statistics were performed beyond the R package(s) utilized. Notably, the supplementary datasets contain statistical values that should be transferred to some capacity into the main text.

Response to comment: Thank you for your suggestion, according to your suggestion, more explanation of how the statistics were performed beyond the R package(s) utilized has been added to the main text. The main analysis parameters of the software used in the analysis have already been provided in the article. (line 88, 101-103)

3. Verbiage that was used to describe findings should be revised (e.g. revising “we got a gene-pathway network” to “we identified…”; “10 genes were finally got”, etc.)

Response to comment: Thank you for your careful evaluation of this manuscript. We revised the title of the manuscript. (line 15, 197, 238) 

Minor comments:

1. Please provide more explanation and description of the methods, including how/why each database was utilized.

Response to comment: We revised the manuscript according to the reviewer’s suggestion. In the portion of Microarray Data from GEO data repository in Materials and methods, why each database was utilized was described: the database was included in the study when it met the four criteria. In the portion of Preprocessing of raw data in Materials and methods, how each database was utilized was described: the selected five gene expression profiles were merged into one file, and then log and baseline transformations were done. More detailed description of other methods has been added to the main text. (line 88, 101-103)

Reviewer #3: Comments:

1. The title of the manuscript should be modified.

Response to comment: Thank you for your careful evaluation of this manuscript. We revised the title of the manuscript. (line 1-4) 

2. If possible graphical abstract should be provided. According to journal format if it is not possible in the main text, it can be provided in the supplementary file for better understanding of the readers (optional).

Response to comment: We adopt your suggestion, we have designed a graphical abstract as supplementary file.

3. The Abstract portion should be modified.

Response to comment: We have modified the Abstract part according to the Reviewer’s suggestion. (line 15-23)

4. Line Numbers should be added throughout manuscript.

Response to comment: We have added the Line Numbers according to the Reviewer’s suggestion.

5. The introduction portion should be rewritten.

Response to comment: Thank you for your suggestion, according to your suggestion, we revised the Introduction portion. (line 28-40, 57-62)

6. The author should give a strong hypothesis why this study in details.

Response to comment: Thank you for your suggestion, we have added some content to explain the aim of our work in the Introduction portion. (line 59-62)

7. Introduction section: (3rd Paragraph) Line no -1: detecte should be corrected.

Response to comment: It has been revised. (line 63)

8. Materials and method section:

Here the authors have simply written they have taken the data and have used the different softwares for the data analysis but the detailed analysis regarding how it is doing that is completely lacking. This should be addressed properly.

Response to comment: We totally agree with your suggestion. As the software and its parameters used in the analysis have already been provided in the article, the data result can be easily reproduced. Moreover, we have provided the main codes used in the article as file “Main_workflow.R”. Additionally, readers can contact the author to fetch the codes.

9. Results section:

All figures need a better resolution. The author should also mention the image source and the software used to make the images.

Response to comment: The high resolution images have been uploaded in Figures folder. The description of drawing software has been added in the main text. (line 88, 101-103)

10. Conclusion section:

The conclusion portion should be rewritten and the author must add a paragraph on future prospects

Response to comment: According to your suggestion, the conclusion portion has been rewritten and a paragraph on future prospects also has been added. (line 257-270)

11. After the conclusion section there should be some small sections like: Supporting information, Acknowledgments, Author Contributions etc

Response to comment: Sections (Supporting information, Acknowledgments, Author Contributions) have been added to the main text. (line 273-303)

12. Correct the PubMed ID of Reference No. 40.

Response to comment: PubMed ID of Reference No. 40 has been corrected. (line 400)

The reviewer specifically raised some concerns regarding the novelty and significance of our work. Based on the reviewers’ comments, we have made extensive alterations to the structure, format, presentation, and analysis of our findings. In response, we have addressed all of the reviewer’s technical concerns with a substantial amount of new data and graphic, supported by additional detailed explanations (the majority of the figures in both the text and supporting information have been revised as requested). If there are any other modifications we could make, we would like very much to modify them and we really appreciate your help. We hope that our manuscript could be considered for publication in Plos One. Thank you very much for your help.

Yours sincerely,

 Yi Zheng

13 Jan, 2024

---

## [Editor Report · Decision Letter 1]

29 Jan 2024

PONE-D-23-26444R1Identification of Potential Biomarkers of Leprosy: A Study Based on GEO DatasetsPLOS ONE

Dear Dr. Zheng,

Thank you for submitting your manuscript to PLOS ONE. After careful consideration, we feel that it has merit but does not fully meet PLOS ONE’s publication criteria as it currently stands. Therefore, we invite you to submit a revised version of the manuscript that addresses the points raised during the review process.

Article is now in shape however needs further efforts to clinical relevance of study in discussion section. Comparative data or evaluation with another studies may give better insight.

Additionally, future prospect with better scope, affected region, populations, policy and technology relevance will connect it for better healthcare management.

We look forward to receiving your revised manuscript.

Kind regards,

Anshuman Mishra, PhD

Academic Editor

PLOS ONE

Journal Requirements:

Additional Editor Comments:

Dear Dr. Yi Zheng,

Thanks for the revision. Article is now in shape however needs further efforts to clinical relevance of study in discussion section. Comparative data or evaluation with another studies may give better insight.

Additionally, future prospect with better scope, affected region, populations, policy and technology relevance will connect it for better healthcare management.

---

## [Author Response · Author response to Decision Letter 1]

15 Mar 2024

We feel great thanks for your professional review work on our article. In response, we have addressed all of the reviewer’s technical concerns. If there are any other modifications we could make, we would like very much to modify them and we really appreciate your help. We hope that our manuscript could be considered for publication in Plos One. Thank you very much for your help.

---

## [Editor Report · Decision Letter 2]

11 Apr 2024

Identification of Potential Biomarkers of Leprosy: A Study Based on GEO Datasets

PONE-D-23-26444R2

Dear Dr. Zheng,

We’re pleased to inform you that your manuscript has been judged scientifically suitable for publication and will be formally accepted for publication once it meets all outstanding technical requirements.

Kind regards,

Anshuman Mishra, PhD

Academic Editor

PLOS ONE
---

## [Editor Report · Acceptance letter]

30 Apr 2024

PONE-D-23-26444R2 

PLOS ONE

Dear Dr. Zheng, 

I'm pleased to inform you that your manuscript has been deemed suitable for publication in PLOS ONE. Congratulations! Your manuscript is now being handed over to our production team.

Kind regards, 

on behalf of

Dr. Anshuman Mishra 

Academic Editor

PLOS ONE